# A Low-Cost, Point-of-Care Test for Confirmation of Nasogastric Tube Placement via Magnetic Field Tracking

**DOI:** 10.3390/s21134491

**Published:** 2021-06-30

**Authors:** Muneaki Miyasaka, Hao Li, Kon Voi Tay, Soo Jay Phee

**Affiliations:** 1School of Mechanical and Aerospace Engineering, Nanyang Technological University, Singapore 639798, Singapore; msjphee@ntu.edu.sg; 2Department of Otorhinolaryngology, Tan Tock Seng Hospital, Singapore 308433, Singapore; li_hao@ttsh.com.sg; 3Department of General Surgery, Woodlands Health Campus, Singapore 069112, Singapore; kon_voi_tay@whc.sg

**Keywords:** magnetic sensors, wearable sensors, real-time systems

## Abstract

In this work, we aim to achieve low-cost real-time tracking for nasogastric tube (NGT) insertion by using a tracking method based on two magnetic sensors. Currently, some electromagnetic (EM) tracking systems used to detect the misinsertion of the NGT are commercially available. While the EM tracking systems can be advantageous over the other conventional methods to confirm the NGT position, their high costs are a factor hindering such systems from wider acceptance in the clinical community. In our approach, a pair of magnetic sensors are used to estimate the location of a permanent magnet embedded at the tip of the NGT. As the cost of the magnet and magnetic sensors is low, the total cost of the system can be less than one-tenth of that of the EM tracking systems. The experimental results exhibited that tracking can be achieved with a root mean square error (RMSE) of 2–5 mm and indicated a great potential for use as a point-of-care test for NGT insertion, to avoid misplacement into the lung and ensure correct placement in the stomach.

## 1. Introduction

The nasogastric tube (NGT) is a flexible rubber or plastic tube which is used for medical purposes such as treatment of ileus or bowel obstruction, stomach lavage, administration of medications, and delivery of nutrients [1]. The NGT is manually inserted through the nose into the stomach often without direct visualization of the location of its tip. Because the tube tip location is unknown, misplacement of the tube is relatively common and it can cause fatal complications [1,2,3]. For instance, there is some possibility of the NGT entering the trachea and the lung instead of the esophagus, resulting in pneumothorax or pneumonia which could be fatal [4].

The pH test is one bedside method to assess the NGT position. The procedure involves aspiration of fluid from the tube and the tube is considered inside the stomach if a pH of 1–5.5 is obtained [5]. However, a misleading pH can be recorded for the patients with hiatus hernia and gastro-oesophageal reflux [4]. Patients taking proton pump inhibitors or requiring continuous enteral feeds could have unexpected pH results [6]. Using an X-ray is an accurate method to confirm the NGT position since the whole length of the tube can be visualized. However, it is reported that misinterpretation of X-rays is the main causal factor of adverse events [5]. The X-ray confirmed tube location is only true at the time the X-ray is taken. If patients have symptoms such as coughing, retching, or vomiting, the X-ray may need to be repeated and there is a small increase of carcinogenic risk. Besides, subsequent clinical procedures are then delayed due to the extra time incurred for patient transfer to the radiology room and for procedures of taking and interpreting X-rays. The delay can take away time crucial for feeding, hydration, and medication [4]. Most importantly, both pH tests and X-ray are only performed post tube insertion. Therefore, harm caused in the midst of the insertion cannot be detected/avoided.

In contrast, electromagnetic (EM) tracking systems, or specifically, electromagnetic sensor guided enteral access systems (EMS-EAS), are able to provide real-time 3D (the anterior and cross-sectional) path of the NGT during the insertion process. Studies showed that 98% of the tube position from EMS-EAS and X-ray corresponded and the remaining 2% can be considered to be caused by the relocation of the tube between EMS-EAS and X-ray, misuse of EMS-EAS system, and misinterpretation of the X-ray [2,7]. Thus, EM tracking can realize a safer and more reliable insertion process and can potentially replace X-ray to reduce the risk of complication, time to next clinical procedure, and cost [8]. However, EM tracking systems are still expensive. For example, CORTRAK EMS-EAS (Avanos Medical, USA) costs £12,000 [9]. The system requires a dedicated single-patient-use tube with a wired sensor head and therefore, the cost for each use is not low.

Passive magnet tracking is another feasible method of NGT tracking. While the EM tracking systems require an active magnetic field generator (external system) and magnetic field receiver (tracking object), the passive magnet tracking system employs external magnetic sensors and a permanent magnet as a tracking target. Various passive magnet tracking methods have been investigated especially in the field of endoscope capsule tracking [10,11,12,13,14]. However, the size of the permanent magnet used for the tracking is usually large and not suitable for NGT insertion. Sun et al. developed a wearable sensor system around the neck to track a permanent magnet embedded at the tip of the NGT [15]. The tube placement is suspected to be incorrect if the sagittal axis position deviates from the expected path because there are distinct position differences between the paths to the esophagus and to the trachea/bronchi. The insertion is considered erroneous if the frontal plane position shifts sideways since the trachea splits laterally into the two primary bronchi. Although the system has a root mean square error (RMSE) of less than 5.3 mm, the accuracy was only verified from −70 to 50 mm from the sensors along the longitudinal axis and the accuracy decreases as the target goes further away from the sensors. Since the average length of the trachea is roughly 100 mm, the tracking up to the bronchi could be difficult to achieve considering anatomical variations [16]. The tracking speed achieved by their system is 10 Hz.

We have previously developed a passive magnet tracking method based on two magnetic sensors, for tracking the magnetically inflatable intragastric balloon capsule (MIBC) [17]. The system uses two three-axis magnetic sensors placed in front and back of the patient. By using the grid search method, the algorithm estimates the position of the magnet embedded in the capsule by searching the sensed magnetic field from a table with a pre-calculated magnetic field. To improve the tracking stability and accuracy, the search range was actively adjusted based on the anatomy of the esophagus and the search threshold was modulated. We demonstrated that the proposed method was able to track the position of the MIBC along the esophagus with a 3.5 mm mean absolute error. However, it could only provide sagittal plane tracking. To detect the deviation of the NGT position toward the lung, frontal plane tracking is required. In addition, for the NGT application, as the magnet size is limited by the NGT size that is smaller than the MIBC size, the tracking becomes more challenging. Therefore, the system from our previous work is not suitable for NGT tracking.

In this work, we aim to achieve inexpensive real-time tracking of NGT insertion, point of care test NGT insertion by implementing a two-sensor-based magnetic tracking method. By introducing the new sensor orientation and modifying the tracking algorithm, we track the frontal plane position of the NGT with a permanent magnet embedded at the tip. By providing real-time visual information of the tip’s position, the user can avoid erroneous insertion of the NGT into the lung and hence, the risks associated with NGT insertion can be reduced. Compared to the EM tracking systems, our approach may not be able to achieve as accurate or as large range tracking due to the nature of the passive magnet tracking method. However, the accuracy and tracking range can be adequate for the NGT tracking application. Since only two sensors are required for tracking, the cost and complexity of the setup is minimal. Additionally, our approach is beneficial because it does not require wiring for the tracking object (permanent magnet). This paper is organized as follows. Section 2 presents the setup of our two-sensor-based tracking system, details of the tracking algorithm, and experimental setup to evaluate our approach. Section 3 presents the results of the experiments followed by the discussion in Section 4. Section 5 provides conclusion and future work.

## 2. Materials and Methods

### 2.1. Two-Sensor Setup

An overview of the two-sensor setup is shown in Figure 1. In this work, we employed two three-axis magnetic sensors with 0.042 μT detectivity (BM1422AGM, ROHM, Japan). The sampling frequency was set at 100 Hz. We used the initial setting of the sensor from the manufacturer and no calibration is carried out. The sensors are placed in front of the patient on the frontal plane with 80 mm separation along the frontal axis without longitudinal offset. Each magnetic sensor is connected to a sensor shield (SHIELD-EVK-001, ROHM, Japan) which is interfaced with an Arduino UNO board (Arduino, Ivrea, Italy). As for the tracking target, a grade N52 neodymium axially-magnetized cylindrical permanent magnet of dimensions Ø 3.18 mm × 9.53 mm (K&J Magnetics, Miami, FL, USA) was embedded at the distal tip of the NGT. The size of the magnet is small enough to fit inside the 14 French gauge (FG) (4.67 mm) or larger NGTs. The tracking algorithm is written in C++ and executed on a laptop PC with the Intel i7-7500U (2 core, 4 thread, and 2.7 GHz) CPU and 16GB RAM. The actual setup can be seen in Section 2.3. The total cost of the system including the PC was about 1000 USD.

Figure 2 shows the anatomy of the digestive and respiratory pathways. The average length of the esophagus (from the cervical esophagus to the EG junction cardia) for an adult is 250 mm [18]. The trachea is 100 mm long on average and it passes vertically down the midline of the chest in healthy subjects and bifurcates at the level of the 4th to 5th thoracic vertebrae [16,19]. This bifurcation is marked by the carina and it corresponds to the palpable landmark of the angle of Louis on the sternum [20]. Left and right main bronchi emerge from the carina. The right the main bronchus is more vertical than the left, making an average angle of 35 (± standard deviation (SD) of 8) degrees with respect to the trachea in a recent study of 2107 Chinese adult patients [21]. The left main bronchus is typically 10 to 15 degrees more horizontal than the right [22]. Both main bronchi further divide into lobar bronchi, and the lobar bronchi divide into segmental bronchi. The luminal diameter of the bronchi decreases as they divide. Because the diameter of a size 14 FG NGT, the typical size used in adult patients, is 4.67 mm, it should likely terminate in a lobar or a segmental bronchus of the right lower lobe [23,24]. Therefore, we are interested in the distance from the carina to the segmental bronchi of the right lower lobe, which is at least 40 mm [19,22] (the sum of the length of the right main bronchus and the bronchus intermedius). Beyond the bronchus intermedius, the length and the diameter of the airways become difficult to predict. Likewise, if the NGT is inserted into the left lung, it will probably pass through the left main bronchus into the left lower lobar bronchus [23,24]. That is a distance of at least 45 mm beyond the carina, the length of the left main bronchus [19,22].

The tracking range covered by the system was ±150 mm in the longitudinal direction from the center of the sensors. Therefore, the esophagus and the airway where the NGT can enter can be covered. In terms of the tracking depth or the tracking in the sagittal axis direction, the sensors can cover up to 100 mm. Since the mean diameter of the anterior–posterior rib cage ranges from 80 to 100 mm, the region of interest is within the tracking range [25].

In the study by Sun et al, the sagittal plane position of the tube below the neck was monitored to differentiate the insertion into the esophagus and trachea/bronchi [15]. However, since there is no reference of where the esophagus or trachea is, it is not always easy to judge misinsertion based on the sagittal plane tracking information. For our approach, since the tracking range is larger, we can focus on the frontal deviation of the NGT position around the chest area to detect incorrect insertion. In addition, our method can confirm whether the NGT passes the esophagogastric junction to the stomach which is beneficial before using the NGT.

### 2.2. Tracking Algorithm

For simplicity, we define the sagittal, frontal, and longitudinal axis as *x*, *y*, and *z* respectively. The algorithm uses the grid search to find the sensor obtained magnetic field from an array of the model-based pre-calculated magnetic field and estimate the position and orientation of the target magnet. The grid search was performed for each of the two sensors individually. In order to isolate the magnetic field of the target magnet from background fields, the magnetic field of the surrounding environment is measured and offset from the sensor value. The sensor-obtained vector after offsetting is denoted as:(1)$$\begin{array}{c} B_{{\mathrm{sensor}}_{j}}=\left[\begin{array}{c} B_{sensor_{j},x}\\ B_{sensor_{j},y}\\ B_{sensor_{j},z} \end{array}\right] \end{array}$$
where the subscript j(=1,2) indicates sensor 1 and sensor 2, respectively and the subscripts *x*, *y*, and *z* indicate the components of the magnetic field vector. Since the magnet has a cylindrical shape, all the positions and orientations can be represented with the five variables (*x*, *y*, and *z* Cartesian coordinates and $\theta_{pitch}$ and $\theta_{roll}$ rotation angles). Therefore, the model-based pre-calculated magnetic field is a function of those five variables and denoted as:(2)$$\begin{array}{c} B_{\mathrm{model}}\left(x,y,z,\theta_{p},\theta_{r}\right)=\left[\begin{array}{c} B_{model,x}\left(x,y,z,\theta_{p},\theta_{r}\right)\\ B_{model,y}\left(x,y,z,\theta_{p},\theta_{r}\right)\\ B_{model,z}\left(x,y,z,\theta_{p},\theta_{r}\right) \end{array}\right] \end{array}$$

However, it is difficult to find the exact same vector due to uncertain factors such as sensor misalignment and noise. To take these uncertainties into account, a search threshold vector $B_{{\mathrm{thresh}}_{j}}={\left[B_{thresh_{j},x},B_{thresh_{j},y},B_{thresh_{j},z}\right]}^{T}$ is introduced. All the sets of the magnet position and orientation that satisfy the below condition are considered as the solutions of the grid search:(3)$$\begin{array}{c} |B_{\mathrm{model}}\left(x,y,z,\theta_{p},\theta_{r}\right)-B_{{\mathrm{sensor}}_{j}}\left|\leq \right|B_{{\mathrm{thresh}}_{j}}| \end{array}$$

The array of the found sets from the grid search is denoted $P_{j}$. Since we have two sensors, two arrays that contain the possible solutions ($P_{1}$ and $P_{2}$) are obtained. The possible solution sets are further narrowed down by extracting the overlapped sets:
(4)$$\begin{array}{l} P_{\mathrm{overlap}12}=\\ P_{1}\cap P_{2}=\left[\begin{array}{ccccc} {\hat{x}}_{1} & {\hat{y}}_{1} & {\hat{z}}_{1} & {\hat{\theta }}_{p,1} & {\hat{\theta }}_{r,1}\\ {\hat{x}}_{2} & {\hat{y}}_{2} & {\hat{z}}_{2} & {\hat{\theta }}_{p,2} & {\hat{\theta }}_{r,2}\\ \vdots & \vdots & \vdots & \vdots & \vdots \\ {\hat{x}}_{N_{12}} & {\hat{y}}_{N_{12}} & {\hat{z}}_{N_{12}} & {\hat{\theta }}_{p,N_{12}} & {\hat{\theta }}_{r,N_{12}} \end{array}\right], \end{array}$$
where $P_{\mathrm{overlap}12}$ indicates the overlapped sets for sensor 1 and sensor 2. $N_{12}$ is the number of the overlapped sets from sensor 1 and sensor 2. Here, we also consider the sensor alignment and noise uncertainties, and the sets from two sensors are considered as overlapped if the difference of all the components are within an overlap threshold. The overlap threshold for sensors 1 and 2 is denoted $P_{\mathrm{thresh}12}$. The final position-tracking result in the frontal plane ($y_{est}$ and $z_{est}$) is obtained by taking the average as follows:(5)$$\begin{array}{ccc} \; & y_{est}=\frac{\sum_{i=1}^{N_{12}}{\hat{y}}_{i}}{N_{12}},\; & z_{est}=\frac{\sum_{i=1}^{N_{12}}{\hat{z}}_{i}}{N_{12}}. \end{array}$$

#### 2.2.1. Precalculation of Magnetic Field

The precalculated magnetic field $B_{\mathrm{model}}\left(x,y,z,\theta_{p},\theta_{r}\right)$ needs to be as accurate as possible to achieve the best localization result. Since the magnetization of the magnet may not exactly match the specification, it is ideal to measure the field around the actual magnet employed rather than using a model. However, since it is time-consuming, we employ the Radia electromagnetic analysis software package from the European Synchrotron Radiation Facility [26,27]. The software uses a boundary integration method and analytical expressions and it can provide more accurate data compared to a simple analytical model and faster computational speed compared to the Finite Element Method. The Radia-generated magnetic field data are utilized for real-time interaction for an electromagnet-based haptic device and a magnetic levitation device that requires an accurate lookup table of magnetic field data [28,29,30,31]. Therefore, the Radia is suitable for our application. The step size for each variable should be decided in accordance with the required tracking accuracy and speed. In this work, we select 10 mm and π/18 for the position and orientation respectively.

#### 2.2.2. Search Range

While a typical grid search uses fixed search ranges, our method utilizes anatomically-constrained search ranges which are updated every computational iteration based on the current state of the target magnet. The entire search range for each variable is limited by the range of the precalculated magnetic field. The global search range for each variable is denoted as $R_{Gx}$, $R_{Gy}$, $R_{Gz}$ for *x*, *y*, and *z* translations, and $R_{Gp}$, $R_{Gr}$ for pitch and roll rotations, respectively (Figure 3). The origin (*O*) of the global coordinate system is located where the trachea and esophagus start. The range of the precalculated magnetic field should include all the possible positions and orientations of the target magnet during the insertion process. In this work, the *x*, *y*, and *z* ranges for each sensor are precalculated from 0 to 200 mm, ±100 mm, and ±150 mm respectively with respect to the center of the sensor. For $\theta_{p}$ and $\theta_{r}$, the magnetic fields are precalculated from 0 to π and 0 to 2π respectively. For the setup with the sensors placed 100 mm below the start of the trachea/esophagus, the global search range for both sensors combined becomes $R_{Gx}=\left[-100,100\right]$ mm, $R_{Gy}=\left[-60,60\right]$ mm, $R_{Gz}=\left[-50,250\right]$ mm, $R_{Gp}=\left[0,\pi \right]$, and $R_{Gr}=\left[0,2\pi \right]$.

If the entire global search range is used for the grid searching for every computational iteration, the localization will be inaccurate since the possible solutions can exist all over the search space. Real-time tracking will fail due to the computational burden. Hence, we introduce the dynamically-constrained instantaneous search ranges which are updated every computational iteration. Although the search ranges for all the variables can be constrained, only *y*, *z* and $\theta_{p}$ are dynamically constrained. This is because the experiments we conducted show unstable tracking when other variables are constrained. The instantaneous search ranges for *y*, *z* and $\theta_{p}$ are denoted as $R_{Iy}$, $R_{Iz}$ and $R_{Ip}$. For *y*, we use
(6)$$\begin{array}{c} R_{Iy}=\left[y_{k}-y_{c},y_{k}+y_{c}\right] \end{array}$$
for the entire global search range. $y_{k}$ is the estimated y position at the *k*th iteration and $y_{c}$ is a constant value to confine the y search range. $R_{Iz}$ and $R_{Ip}$ are divided into three regions (region 0, 1, and 2) based on the positions along the longitudinal axis. The details of each region are explained as follows.

Region0: This region is used until the first estimation is provided. The target magnet will be out of the sensor’s sensible range at the beginning of the NGT insertion. As the insertion progresses, the tip of the tube will enter the search range always from the top. Therefore, we need to focus only on the top slice of the entire search range. Since the insertion speed and the anatomy or tilt angle of the esophagus and trachea are limited, the search ranges can be constrained as follows:
(7)$$\begin{array}{cc} \; & R_{Iz}=\left[z_{top},z_{top}+v_{max}dt\right]\in \left[z_{top},z_{0}\right] \end{array}$$(8)$$\begin{array}{cc} \; & R_{Ip}=\left[0,\theta_{p,max}\right], \end{array}$$
where $z_{top}$ is the top end of the z search range, $v_{max}$ is the possible maximum speed of tube insertion, dt is the time step in between the iterations, $z_{0}$ is the first estimation of z position, and $\theta_{p,max}$ is the possible maximum pitch angle of the tube tip inside the esophagus and trachea.Region1: This region covers from the first estimated z position to the level of the carina or end of the trachea. The search ranges are constrained based on the current position, orientation, speed of the tube tip and the possible tilt angle of the tube tip inside the esophagus and the trachea.
(9)$$\begin{array}{cc} \; & R_{Iz}=\left[z_{k}-v_{max}dt-z_{unc},z_{k}+v_{max}dt+z_{unc}\right]\in \left(z_{0},z_{car}\;\right] \end{array}$$(10)$$\begin{array}{cc} \; & R_{Ip}=\left[\theta_{p,k}-\phi_{R1}-\theta_{p,unc},\theta_{p,k}+\phi_{R1}+\theta_{p,unc}\right]\in \left[0,\theta_{p,max}\right] \end{array}$$
where $z_{k}$ and $\theta_{p,k}$ are the estimated z position and pitch angle at the *k*th iteration. $z_{car}$ is the z position of the carina. $z_{unc}$ and $\theta_{p,unc}$ are the uncertainties from the estimation errors. $\phi_{R1}$ is the possible pitch rotation during one computational iteration inside the esophagus and trachea.Region2: This region covers from the end of region 1 to the bottom end of the search range. The tube tip could be inside the trachea, bronchi, esophagus, or stomach. Therefore, any pitch angle is possible.
(11)$$\begin{array}{cc} \; & R_{Iz}=\left[z_{k}-v_{max}dt-z_{unc},z_{k}+v_{max}dt+z_{unc}\right]\in \left(z_{car},z_{bottom}\right] \end{array}$$(12)$$\begin{array}{cc} \; & R_{Ip}=\left[\theta_{p,k}-\phi_{R2}-\theta_{p,unc},\theta_{p,k}+\phi_{R2}+\theta_{p,unc}\right]\in \left[0,\pi \right], \end{array}$$
where $z_{bottom}$ is the bottom end of z search range and $\phi_{R2}$ the possible pitch rotation during one computational iteration inside the trachea, bronchi, esophagus and stomach.

#### 2.2.3. Threshold Modulation

The result of the grid search is significantly affected by the search threshold $B_{{\mathrm{thresh}}_{j}}$ which is used in the search condition in Equation (Equation 3). For each sensor, $n_{j}$ sets of solutions are found from the grid search but if the found sets are too many or too few, the following process of finding the overlap becomes challenging. Real-time tracking will fail if $n_{j}$ is too large and tracking itself will fail if $n_{j}$ is too small. It is found that $n_{j}$ is dependent on the z position of the target magnet and the threshold is dependent on the magnitude of $B_{{\mathrm{sensor}}_{j}}$. Therefore, $B_{{\mathrm{thresh}}_{j},k+1}$ ($B_{{\mathrm{thresh}}_{j}}$ at k+1th iteration) is controlled to obtain the right amount of solution sets using the following proportional derivative (PD) controller type modulator:(13)$$\begin{array}{cc} \; & B_{{\mathrm{thresh}}_{j},k+1}=B_{{\mathrm{thresh}}_{j},k}+K_{p_{j},k}\left(z\right)e_{j,k}+K_{d_{j},k}\frac{de_{j,k}}{dt}\\ \; & where \end{array}$$(14)$$\begin{array}{cc} \; & K_{p_{j},k}\left(z\right)=\alpha \left(z\right)\left|B_{{\mathrm{sensor}}_{j},k}\right| \end{array}$$(15)$$\begin{array}{cc} \; & K_{d_{j},k}=\beta \left|B_{{\mathrm{sensor}}_{j},k}\right| \end{array}$$(16)$$\begin{array}{cc} \; & e_{j,k}=n_{target}-n_{j,k} \end{array}$$(17)$$\begin{array}{cc} \; & \frac{de_{j,k}}{dt}=\frac{e_{j,k}-e_{j,k-1}}{t_{k}-t_{k-1}} \end{array}$$

$B_{{\mathrm{thresh}}_{j},k}$ is the search threshold at the *k*-th iteration. $K_{p_{j},k}\left(z\right)$ is the proportional gain vector and $K_{d_{j},k}$ is the derivative gain vector at the *k*th iteration. α(z) and β are the modulation coefficients for the proportional gain and for the derivative gain respectively. $e_{j,k}$ represents the error between the number of the targeted sets $n_{target}$ and the detected sets at the *k*th iteration $n_{j,k}$. The time at the k−1th and *k*th iteration are denoted $t_{k-1}$ and $t_{k}$ respectively.

In addition, the overlap threshold $P_{\mathrm{thresh}12}$ directly affects the computation of the final estimation. Depending on the values of the overlap threshold, no overlap could be detected even if the grid search results from the pairs of the sensors contain virtually identical values. Real-time tracking could fail if too many overlaps are found. The overlap threshold contains five components; $x,y,z,\theta_{p},$ and $\theta_{r}$.
(18)$$\begin{array}{cc} \; & P_{\mathrm{thresh}12}=\left[\begin{array}{ccccc} P_{thresh12,x}\;\;\; & \;\;P_{thresh12,y}\;\;\; & \;\;P_{thresh12,z}\;\;\; & \;\;P_{thresh12,p}\;\;\; & \;\;P_{thresh12,r} \end{array}\right] \end{array}$$

$P_{\mathrm{thresh}12}$ is updated at every iteration based on the values of $z_{est}$, $n_{1}$, $n_{2}$, and $N_{12}$. Details (i.e., how and what values are selected) are described in the following section.

### 2.3. Experimental Setup

We perform two different experiments to investigate the performance of the two-sensor-based magnetic tracking system for NGT. In the first experiment, the position tracking accuracy is evaluated. The magnet is placed at various locations on the frontal planes with *L* (distance from the sensors to the magnet along the sagittal axis) = 80 and 100 mm. The reference (ground truth) position of the magnet was measured/set using manual linear sliders with 1 mm marks. The experimental setup and the locations tested in the frontal plane are illustrated in Figure 4. The tracking was run for 5 s (which results in roughly 250 to 400 data points) for each location. Assuming the sensors are placed on the chest skin 100 mm below the start of the trachea/esophagus, this experiment can check the tracking accuracy within the esophagus and airway where the NGT can enter.

In the second experiment, we used a mock-up esophagus and airway constructed using silicone tubes (Figure 5). The inner diameters of the tubes for the esophagus, trachea, main bronchi, and lobar/segmental bronchi are selected to be 20, 20, 15, and 10 mm respectively based on the anatomy of the average adult, and the other dimensions are set as indicated in Figure 2 [21,32,33,34]. The NGT with the permanent magnet is inserted into each pathway separately to examine whether each tracking result is distinct enough for reliable misinsertion detection. To show the variation of the tracking performance, the NGT is inserted into each pathway three times. The experiment is performed for *L* = 80 and 100 mm. For both experimental setups, the materials used near the sensors and magnet are all non-magnetic. Although some magnetic materials are used in the linear sliders for position adjustment, they are sufficiently distanced from the tracking range and have no effect on the tracking performance. To smooth out the short-term fluctuations or outliers in the raw estimation result, a moving average filter was applied with a fixed subset size of 5.

The parameters for the instantaneous search ranges used for the experiment are summarized in Table 1. We use $z_{top}$ = 0 and $z_{bottom}$ = 250 mm considering the global search range described in Section 2.2.2. $z_{car}$ is determined based on the average anatomy and *d* is selected arbitrarily. The average NGT insertion time with and without the CORTRAK EMS-EAS is reported to be 28.8 s and 369.6 s respectively, while the average NGT insertion length is up to 730 mm [35,36]. Therefore, the average insertion speed can be as fast as 25 mm/s. To ensure the tracking with sudden and unexpectedly high insertion speed, $v_{max}$ is selected to be four times larger. For $\phi_{R1}$ and $\phi_{R2}$, since there is no literature regarding the rotational speed of the tube tip, the values are empirically determined. $z_{unc}$ and $\theta_{p,unc}$ are set to be the same as the step size of the precalculated magnetic field.

For the PD modulator, $n_{target}$ is selected to be 500 to balance the computational time and the tracking accuracy. Then, the parameters for the PD modulator are adjusted manually as experiments are carried out until the output from the modulator stays near the target value. α(z) is shown in Table 2 and β is determined to be 0.01. The value for $P_{\mathrm{thresh}12}$ is also adjusted manually such that $N_{12}$ always becomes approximately 500. The values and conditions are summarized in Table 3. When multiple conditions are met simultaneously, the condition toward the top of the list is prioritized.

## 3. Results

The result of testing tracking accuracy are presented in Figure 6. The average estimated magnet position and the SD for each axis are plotted together with the true magnet position. For most of the positions, the estimation is very close to the actual position of the magnet except when z = 250 mm where z and y position errors appear to suddenly increase. The y and z-axis errors along the z position are plotted in Figure 7. While there is no obvious trend of y position error change along z-axis, z-position error suddenly increases for z = 250 mm. This could be because the magnet is too far from the sensors and the magnetic field is too small to provide an accurate estimation. Table 4 shows the RMSE of y and z position as well as the RMSE when the result of z = 250 mm is excluded. For both L = 80 mm and 100 mm, we observed decent y tracking accuracy with below 3 mm RMSE. When the average of all the experiments are taken, z position RMSE for L = 80 and 100 mm are 3.55 mm and 5.50 mm. On the contrary, when disregarding the result of z = 250 mm, the z tracking RMSE becomes less than 2 mm.

The result of the tracking when NGT is inserted into the dimensionally correct mock-up esophagus and airways is shown in Figure 8 and Figure 9 (for L = 80 and 100 mm respectively). Since the NGT is smaller than the size of the mock-up esophagus and airways, the insertion path is different for every trial. It is observed that the tracking for all the trials stays mostly within the vertical dashed lines which indicate the inner diameter of the esophagus when the NGT is inserted into the esophagus. When the NGT goes into the airways, we observe large deviations of the y position which are right below the sensor level (*z* = 100 mm) and outside the vertical dashed lines. As this deviation is very distinct, it can be used to determine the erroneous insertion. Although the tracking may go beyond the vertical dashed lines for esophagus insertion, it only happens near *z* = 250 mm and therefore, it should not be considered as misinsertion.

## 4. Discussion

In this work, a laptop PC with a relatively high-performance CPU is used. The NDI Aurora system has a 40 Hz update rate which is considered adequate for real-time tracking [15]. Since the current system achieved a tracking frequency from 50 to 70 Hz, there is a potential to meet the requirement of real-time tracking by using even lower-performing, lower-cost PCs.

The experiments indicated the z-axis tracking accuracy decreased below z = 250 mm. Although the magnitude is different, a similar trend was observed for both L = 80 and 100 mm. If the error increases in a spatially consistent manner, the error might be compensated. We plan to use a neural network or machine learning to identify the spatial characteristic of the error along the z-axis.

The experiments were performed without biological tissue in between the sensors and the NGT. This is because it was found that the tracking performance will not be affected by the biological tissue in the study we previously carried out [17]. The magnetic field generated by live humans ranges from 20 fT to 1 nT [37]. Therefore, it is out of the sensible range of the sensor we employed and it will not affect the tracking performance.

Metal implants may affect the tracking performance depending on the material, size, and location. Therefore, we will investigate to determine whether the system can be appliable for patients with implants.

In this work, the system was tested at room temperature (25 °C). For the real application, the magnet’s temperature will increase due to the body core temperature which is about 37 °C. Since the strength of the magnetic field is dependent on temperature, the effect of the temperature on the tracking performance needs to be investigated.

Our system is compared to the one developed in [15] side-by-side in Table 5. The number of sensors required for our system is about one-fifth of that in the latter system. Since the tracking volume is not specifically stated in their work, it was deduced from the experimental setup and results. Although the tracking volume may be similar, coverage of the airway beyond the main bronchi or the entire esophagus is lacking, due to its placement at the neck. The RMSE of our system is slightly worse when considering the entire range but excluding *z* = 250 mm, our system may outperform their system.

One limitation is our approach will not work when the magnet/NGT is already inside the tracking range as the initialization or offsetting of the background magnetic field is unable to be performed. This means the presented approach is not capable of confirming the location of the already inserted NGT which is necessary for patients with NGT for the long term. In addition, since the magnet is embedded inside the NGT, the patients with these NGT are not able to take MRI scans when necessary. Therefore, it would be advantageous to make the magnet removable and re-insertable.

The other limitation is that no tracking is available beyond the EG junction (cardia). This is enough to fulfill the goals of this work, but providing tracking for the rest of the stomach can be useful in confirming the final location of the NGT, for increased patient safety. Furthermore, when the tubes need to be delivered to the deeper GI tract such as nasoduodenal and nasojejunal intubations, the tracking range needs to be amplified further. It can be done by employing more accurate sensors and a stronger permanent magnet (i.e., N55 grade) or another set of sensors placed around the stomach area.

## 5. Conclusions

This work presents a low-cost, two-sensor-based magnetic tracking system for NGT insertion. Using the grid search combined with a dynamically adjusted search range and PD threshold modulator, we achieved real-time tracking of a magnet embedded at the tip of the NGT. The tracking accuracy and range obtained from the experiments indicate that the presented system is capable of detecting the deviation of the position in the frontal plane to determine when the tube wrongly progresses into the bronchi.

To move forward, the system needs to be tested in humans. For safety, it is ideal to make the magnet removable after NGT insertion. Therefore, a magnet should be connected to a guidewire instead of embedding in the NGT. The coating/material of the magnet and the guidewire should be carefully selected to ensure biocompatibility with and resistance against gastric acid.

Although this work focuses on the application for NGT insertion, the proposed system could be applied for tracking the MIBC, endoscopic capsules, and other medical devices [38,39,40].

## Figures and Tables

**Figure 1 sensors-21-04491-f001:**
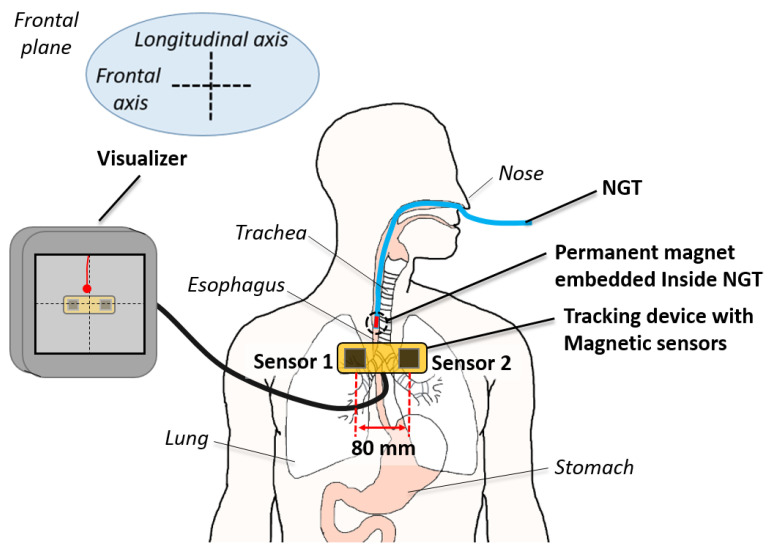
Overview of the two-sensor-based magnetic NGT tracking system.

**Figure 2 sensors-21-04491-f002:**
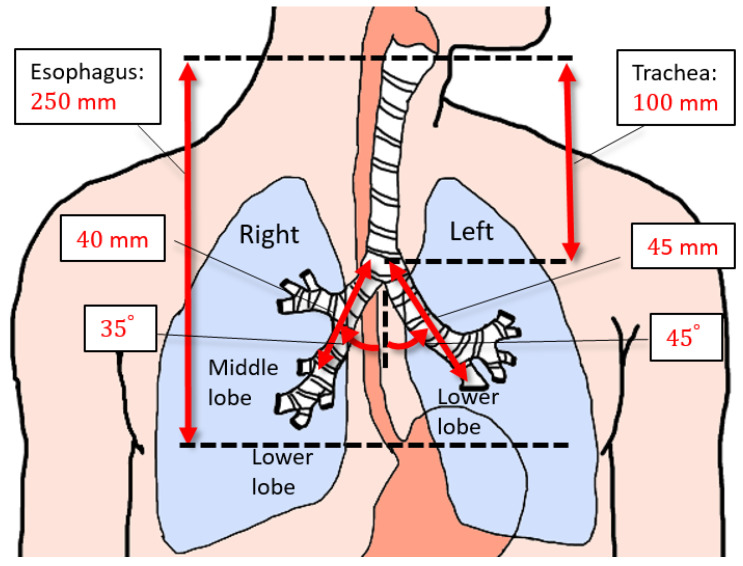
Dimensions of the esophagus and airway where the NGT can enter.

**Figure 3 sensors-21-04491-f003:**
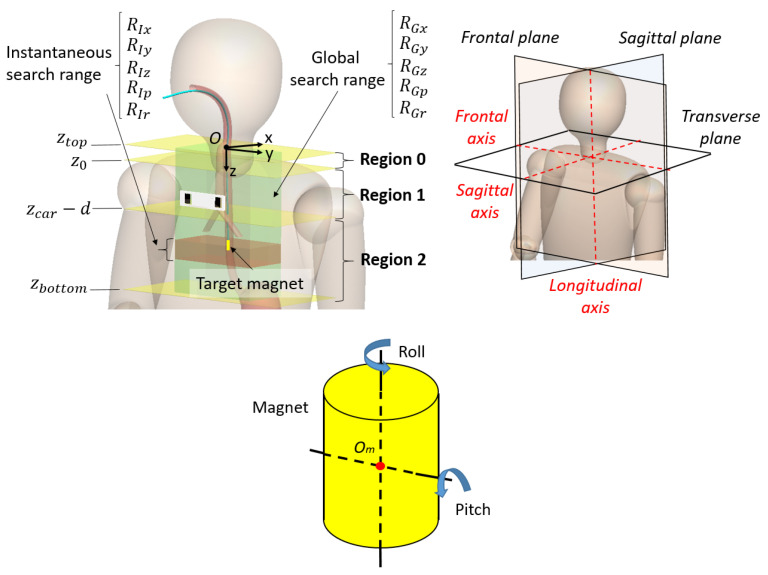
Top: Illustration of the search regions, global search range, and instantaneous search range. Bottom: Definition of pitch and roll rotations and the origin of the magnet ($O_{m}$).

**Figure 4 sensors-21-04491-f004:**
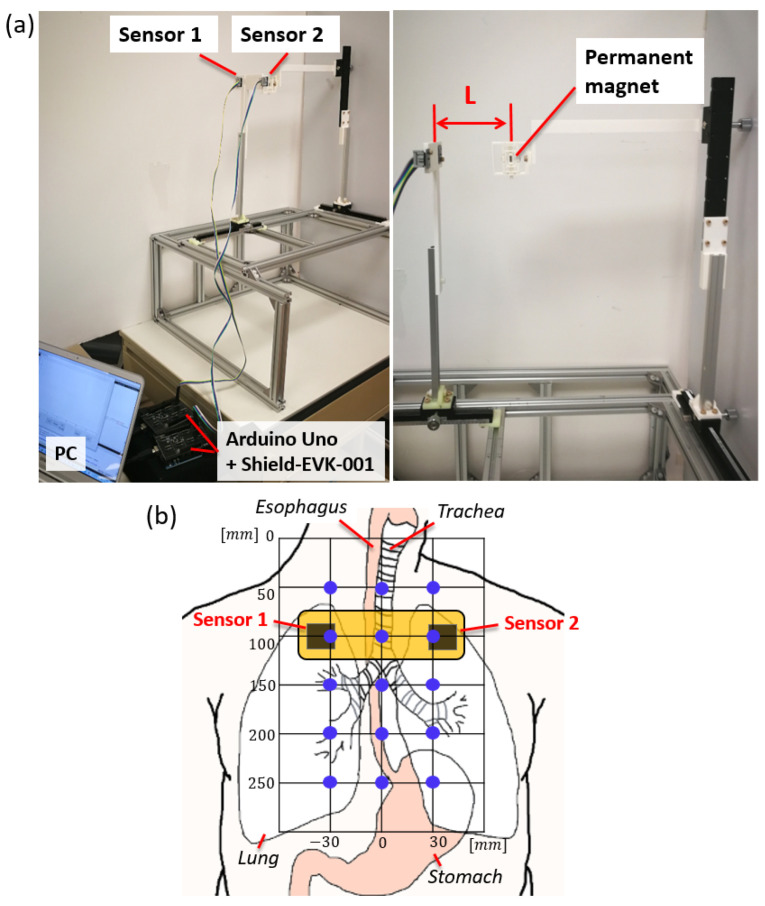
(**a**) The first experimental setup to test the position tracking accuracy. (**b**) The tracking accuracy was evaluated at the 15 locations indicated in blue dots.

**Figure 5 sensors-21-04491-f005:**
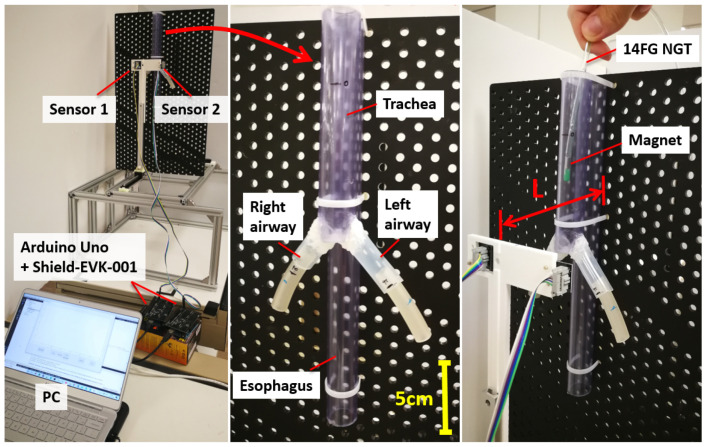
Mock-up esophagus and airway for the second experiment. The 14 FG NGT with a magnet is manually inserted into each path (esophagus, right airway, and left airway).

**Figure 6 sensors-21-04491-f006:**
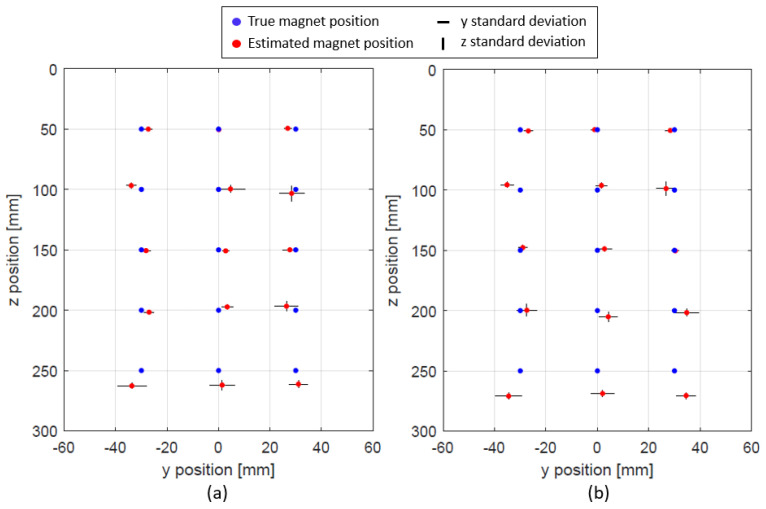
The average y and z position error and SD for L = (**a**) 80 mm and (**b**) 100 mm.

**Figure 7 sensors-21-04491-f007:**
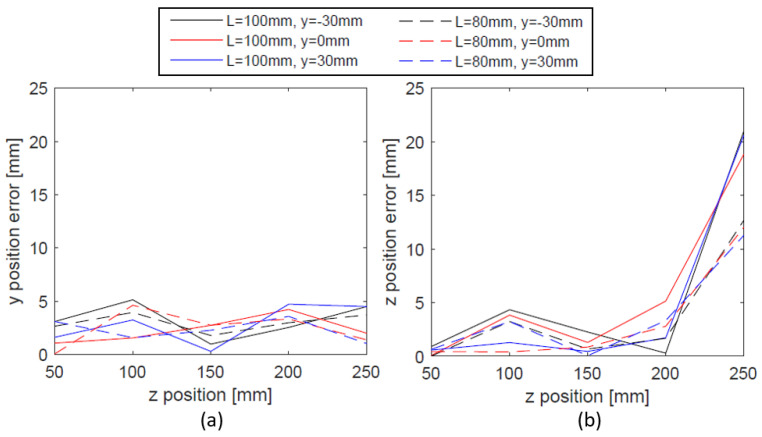
The average position error and SD along z-axis for (**a**) y position and (**b**) z-position. The results for both L = 80 and 100 mm are shown.

**Figure 8 sensors-21-04491-f008:**
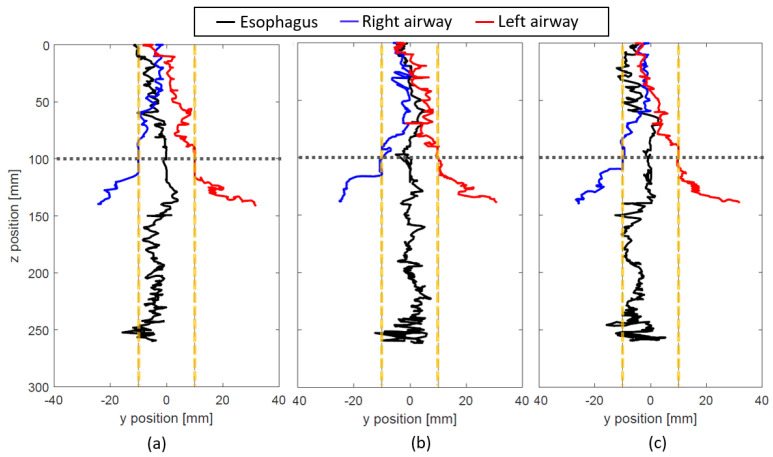
Tracking experiment with the dimensionally accurate mock-up esophagus and airway when L = 80 mm. Three trials of inserting the NGT into each path (the esophagus (black), right airway (blue), and left airway (red)) are shown in (**a**–**c**). The inner diameter of the esophagus and trachea is represented by the yellow vertical dashed lines. The horizontal dashed line indicates the vertical location of the sensors.

**Figure 9 sensors-21-04491-f009:**
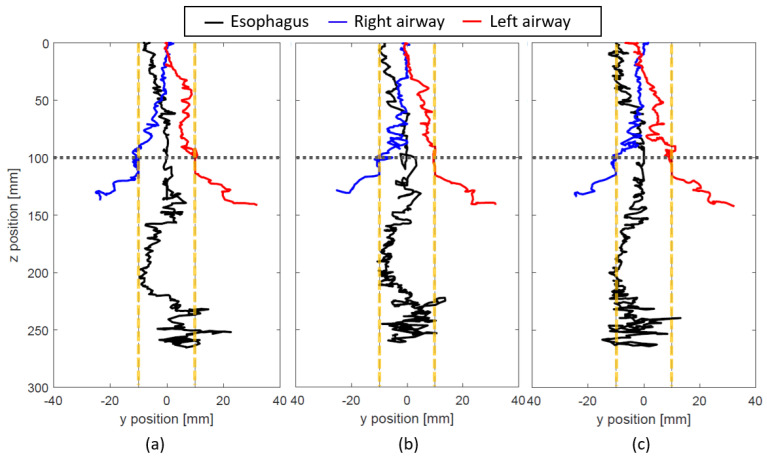
Tracking experiment with the dimensionally accurate mock-up esophagus and airway when L = 100 mm. Three trials of inserting the NGT into each path (the esophagus (black), right airway (blue), and left airway (red)) are shown in (**a**–**c**). The inner diameter of the esophagus and trachea is represented by the yellow vertical dashed lines. The horizontal dashed line indicates the vertical location of the sensors.

**Table 1 sensors-21-04491-t001:** The parameters for the instantaneous search range.

Parameter	Value
$z_{top}$	0 mm
$z_{bottom}$	250 mm
$z_{car}$	100 mm
$y_{c}$	80 mm
$v_{max}$	100 mm/s
$\theta_{p,max1}$	5π/18
$\phi_{R1}$	π/18
$\phi_{R2}$	π/9
$z_{unc}$	10 mm
$\theta_{p,unc}$	π/18

**Table 2 sensors-21-04491-t002:** Value of α that depends on the range of $z_{est}$.

$z_{est}\left[\mathrm{mm}\right]$	$\alpha (z_{\mathrm{est}})$
0	120
(0, 20]	100
(20, 80]	70
(80, 120]	80
(120, 180]	40
(180, 220]	80
(220, 290]	90
greater than 290	100

**Table 3 sensors-21-04491-t003:** Conditions for selecting $P_{\mathrm{thresh}12}$ and the corresponding values for each component. The conditions toward the top of the list are prioritized when multiple conditions are met.

	$P_{\mathrm{thresh}12}$	
Condition	*x*, *y*, *z*, and $\theta_{p}$	$\theta_{r}$
	Components	Component
0 ≤ $z_{est}$ < 2.5	0.4	0.8
$n_{1}$ or $n_{2}$ < 100	1.25	2.5
$n_{1}$ and $n_{2}$ < 500	1	2
500 < $N_{12}$	0.5	1
100 < $N_{12}$ ≤ 500	0.75	1.5
50 < $N_{12}$ ≤ 100	1	2
0 < $N_{12}$ ≤ 50	1.75	3.5
$N_{12}$= 0	2.5	5

**Table 4 sensors-21-04491-t004:** RMSE for L = 80 and 100 mm.

		*y*	*z*	*y*	*z*
		[mm]	[mm]	[mm]	[mm]
				(Excluding *z* = 250 mm)	
L [mm]	80	2.60	3.55	2.47	1.44
	100	2.83	5.50	2.62	1.84

**Table 5 sensors-21-04491-t005:** Comparison of the tracking performance of the proposed and the other magnetic sensor-based tracking systems. For our approach, the RMSE when *z* = 250 mm is excluded is in the parenthesis.

	Proposed	Ref [15]
# of sensors	2	11
Volume [mm^3^]	100 × 120 × 250	100 × 100 × 200
Magnet size [mm]	Φ3.2 × *L*9.6	Φ3.2 × *L*9.6
RMSE in 1D [mm]	2.60 − 5.50	0.92 − 4.05
	(1.44 − 2.62)	

## Data Availability

The data presented in this study are available on request from the corresponding author.

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
