# Peer review of "A Low-Cost, Point-of-Care Test for Confirmation of Nasogastric Tube Placement via Magnetic Field Tracking"

_sensors, 2021, doi:10.3390/s21134491_

Round 1

Reviewer 1 Report

This paper describes the experimental implementation of a nasogastric tube insertion tracking system based on magnetic field sensors. The topic is worthy of investigation, the manuscript is sound and well structured and the results are reliable.

In my opinion, the major weakness of the paper is the tracking algorithm. The calibration part of the process is a bit obscure. A grid scanning is considered, but then FEM calculations are used. Finally, few (15) points are experimentally measured (I suppose that with a fixed magnet orientation). Has been considered an artificial neural networks approach? I guess that, at the price of getting a good training set, results should notably improve. Please, discuss this.

In addition, the number of sensors (2) seems to be arbitrary and unnecessarily small (from the economic and sampling rate points of view). Why not using an array with, for example, 4 or 8 sensors? 

Other minor comments:

Writing "0.042 µT sensitivity" is a misinterpretation. It is better to use "detectivity" 

Some grammar errors:

"The experimental results exhibited the tracking can be achieved"

" it is reported misinterpretation of X-rays is the main
33 causal factor"

"We demonstrated the proposed method was able"

Please, check again the manuscript.

Ref [1] is incomplete.

Reviewer 2 Report

The manuscript proposed a method for real-time tracking of nasogastric tube (NGT) for inserting it into stomach. The tracking system is based on two magnetic sensors. Magnetic field intensity of each sensor is simulated and compared with the sensor reading to estimate the location of a permanent magnet embedded inside the NGT. The electromagnetic tracking system is able to detect the location of the NGT in real time within an error of 2-5mm, thus it can help avoiding misplacement of the tube. I would recommend its publication on Sensors with following minor revisions.

  1. The cost of the real-time tracking system should be included.
  2. Are the simulation results by Radia electromagnetic analysis software package from the European Synchrotron Radiation Facility preloaded in the computer of the tracking system? If so, what is the minimum requirement of the computer CPU? What is the tracking frequency required by clinic?
  3. In clinic use, will some patients with metal parts implanted still be able and be safe to use this tracking system?
  4. Please elaborate the process for trial and error determination of the sensor parameters for alpha, beta, and Pthresh12 in Table 2 and Table 3.
  5. In equation 11, what is the d?
  6. In Figure 5, what is the material of the plastic tube?

Reviewer 3 Report

The paper presents results about the development of a magnetic tracking system for nanogastric tube. The paper is well structured and the targeted application is highly relevant.

There are several aspects of the design that shall be discussed:

The authors have chosen the approach of integrating a permanent magnet at the tip of the tube. This solution is attractive for its simplicity especially considering the integration of the magnet into the nanogastric tube. However, the authors should explain more in details the pros and cons of integrating a magnet within the tube vs. integrating a magnetic sensor within the tube (and generate a field from an external source for tracking purposes)

An estimation of the temperature coefficient of the magnet is needed. Magnets are strongly temperature dependent. In the application, the temperature will increase significantly during insertion into the human body.

A analysis of the influence of repeatability between magnet samples should also be carried on.

There are no comments about the sensor calibration. Was a calibration of the sensor required? The offset is cancelled through an auto-zero procedure which is a good strategy. However, the gain error of the sensor (error an its sensitivity) is not addressed.

Furthermore, the paper has some shortcomings especially about the errors calculations. The paper presents a position and orientation measuring system. In this context the calculation of the measurement uncertainty shall be carried on (according to ISOGUM). The most important source of uncertainty shall be given. The authors provide some error estimation without any description of the measurement of the reference values. This is the case for instance in the second experiment where a value of RMSE is given without any explanation about the measurement of the true values of the positions and orientations.

It is unclear while looking at Fig 5 how the y and z positions can be accurately estimated without any control of L (which corresponds to the x coordinate). The NGT is inserted manually through a tube of 20 mm diameter and x will vary. Does it affects the estimation of y and z? The field is significantly affected at the sensor location when L is varying.

The authors should indicate how many points are used for the averaging of the data. In the 1st experiment (static) 250 to 400 points are used. This cannot be the case during the second experiment (dynamic) considering the acquisition rate of 50 to 70 Hz which is given.

The authors should add in the discussion some comments about the capability of the system to adapt to strong changes of L. For instance after a reflux, would the threshold value be still valid?

Please review this sentence:« the magnet except when z = 250 mm where z position error and y appears to suddenly increase »

Reviewer 4 Report

Remarks:

According to the datasheet the magnetic sensitivity of BM1422AGM is 0.042μT. What are the the lowest precalculated field values? Especially in the z>250mm range where the accuracy of the system significantly decreases.

Pitch and roll angles as well as their zero positions were not defined clearly.

In equation (3) an absolute value brackets is missing on the right hand side (in my opinion).

For threshold modulation a proportional-derivative (PD) algorithm was used. What is the background of this controlling strategy? Why is this better than P or PI controls? Please, explain or add reference.

Formula (13):  

Bthreshj,k+1=Bthreshj,k+Kpj,k(z)*ej,k+Kdj,k*dej,k/dt

If it is a PD algorithm why Bthreshj,k is present on the right-hand side? It is an integration. Please, resolve this contradiction.

Reviewer 5 Report

The work entitled, ' A Low Cost, Point of Care Test for Nasogastric Tube Insertion via Magnetic Field Tracking' is a new work presented by  Miyasaka et al. The findings are new and can be published after few minor corrections;

  1. The abstract could be briefed.
  2. ' Passive magnet tracking' is not clearly discussed. It will be better if the author provide some more information in introduction section.
  3. For Fig. 5. make one scale bar to assume the machine length and actual dimension.

Round 2

Reviewer 3 Report

The authors have increased the quality of their paper following the reviewers recommendations.

The temperature coefficient of the magnet (ppm/°C) should be available in its datasheet. This value should be given in the paper.